# Self-Powered Galvanic Vibration Sensor

**DOI:** 10.3390/mi13040530

**Published:** 2022-03-27

**Authors:** Yik-Kin Cheung, Zuofeng Zhao, Hongyu Yu

**Affiliations:** 1Department of Mechanical and Aerospace Engineering, The Hong Kong University of Science and Technology, Hong Kong, China; ykcheungab@connect.ust.hk; 2School of Earth and Space Exploration, Arizona State University, Tempe, AZ 85281, USA; zuofeng.zhao@asu.edu

**Keywords:** vibration, self-powered vibration sensor, electrochemical vibration sensor, battery

## Abstract

The development of the IoT demands small, durable, remote sensing systems that have energy harvesters and storage. Various energy harvesters are developed, including piezoelectric, triboelectric, electromagnetic, and reverse-electrowetting-on-dielectric. However, integrating energy storage and sensing functionality receives little attention. This paper presents an electrochemical vibration sensor with a galvanic cell (Zn-Cu cell) as energy storage and a vibration transducer. The frequency response, scale factor, long-term response, impedance study, and discharge characteristics are given. This study proved the possibility of integrating energy storage and vibration sensing functionality with promising performance. The performance of the sensor halved within 74 min. The longevity of the sensor is short due to the spontaneous reactions and ions drained. The sensitivity can be restored after refilling the electrolyte. The sensor could be rechargeable by changing to a reversible electrochemical system such as a lead–acid cell in the future.

## 1. Introduction

The development of the IoT demands a small remote sensing system to operate long-term without battery replacements. Other than minimizing power consumption, one approach is energy harvesting from the environment to power the readout and transmission circuit. The common vibration energy harvesting mechanism includes piezoelectricity [1,2,3,4,5,6,7], triboelectricity [8,9,10,11,12], electromagnetic [13], and reverse-electrowetting-on-dielectric [14,15,16]. They transduce mechanical energy into electricity directly and are suitable for self-powered sensing applications.

Piezoelectric material generates an electric field with applied mechanical stress and is mature in vibration monitoring applications. The piezoelectric nanogenerator (PENG) combines nanostructure and piezoelectric material to harvest ubiquitous mechanical energy in the environment [4]. It offers a superior operational lifetime of up to 20 million cycles while maintaining mechanical and electrical stability [3]. However, the material choice is limited and costly. Therefore, novel materials such as cellulose and low-dimensional material composites are under development [6].

The triboelectric nanogenerator (TENG) is emerging due to various material choices, simple fabrication, simple structure, and low cost. It has potential in many self-powered sensors and human–machine interfaces [10]. Researchers have developed a green TENG based on cellulose material derived from wood with an energy harvesting efficiency of 1600 V/N⋅m^2^ that can sustain 100,000 cycles [11]. However, TENG has a short operational life due to repetitive mechanical contact and sliding during energy generation. The most durable TENG reported can sustain 300,000 cycles [12], two orders of magnitude less than the PENG. There are also hybrid piezo-triboelectric nanogenerators (HNG) [17] to improve power output further.

In addition to PENG and TENG, the electromagnetic generator [13] can have a high output power of 27 mW using a coil and magnet. However, it is difficult to miniaturize and integrate into a small system. The reverse electrowetting of dielectric has no solid contact during energy generation and is free from longevity, but the power density is tiny (53.3 nW/cm^2^) [16].

Energy harvesting depends entirely on the environment, and energy storage can improve versatility. On the other hand, there is little attention on integrating energy storage and sensing functionality to become smart energy storage. Combining smart energy storages and energy harvesters could result in a multimodal, compact, remote sensing system. This paper presents a novel concept to use a simple galvanic cell as a self-powered vibration sensor.

### Sensing Principle

The molecular electronic transducer (MET) is a category of electrochemical motion sensors that consists of a liquid electrolyte as proof mass. It senses motion based on enhancing mass transport by convection, which leads to a varying reaction rate at the electrodes under a constant bias potential [17]. This principle has been applied to linear accelerometers [18,19,20] with a straight channel and angular accelerometers [21] with a circular channel. Recently, it extended to tilt sensors [22] with a partially filled circular channel and sensing coverage area instead of flow rate. This mechanism has a high signal-to-noise ratio, structural robustness, and low power consumption. A conventional MET is an electrolytic cell that requires a potential across the anode and cathode. On the other hand, a galvanic cell stores energy and can adopt a similar sensing mechanism, resulting in a self-powered galvanic vibration sensor.

Referring to Figure 1a, the sensor consists of a rectangular channel with two elastic membranes at both ends. The copper and zinc electrodes fixed at the bottom wall immerse in the saturated copper (I) iodide (CuI) solution. A spontaneous displacement reaction between zinc electrodes and electrolyte forms a diffusive layer (grey color) with zinc ions (Figure 1b), forming two half cells, $\mathrm{Zn}/{\mathrm{ZnI}}_{2}$ and Cu/CuI, respectively. Connecting to an external load (i.e., readout circuit), this drives oxidation in the anode (Zn electrode) and reduction in the cathode (Cu electrode). The overall reaction has a standard potential of 1.28 V:$${\mathrm{Zn}}_{\left(s\right)}+2{\mathrm{CuI}}_{\left(\mathrm{aq}\right)}\to {\mathrm{ZnI}}_{2\left(\mathrm{aq}\right)}+2{\mathrm{Cu}}_{\left(s\right)}$$

During vibration, the bulk electrolyte moves opposite to the acceleration due to inertia and deforms the membranes (Figure 1a). The flow accompanied by the motion disturbs the diffusive layer (Figure 1c) and mass transport near the electrodes, resulting in the variation in the reaction current under a constant load and can convert to the physical quantity.

## 2. Materials and Methods

### 2.1. Fabrication and Structure

Referring to Figure 2, we first prepare the elastomer membranes: mix the silicone parts A and B in a 1:1 ratio (Smooth-On Ecoflex 00 30) and stir thoroughly. Degas and pour the mixture into a Petri dish without spreading to the edge. Cover the dish with aluminum foil and place it on a level surface to cure for 4 h at room temperature. The cured Ecoflex membrane is carefully peeled off and trimmed into the appropriate size. Repeat the above processes to obtain two membranes. Then, the two sheets are fixed to a 3D-printed housing (Formlab, clear resin with Form 3 SLA printer) using clamp plates, screws, and nuts to seal the two ends of the rectangular channel. Insert a 0.5 mm diameter copper and zinc wire into the rectangular channel via the inserting holes on the housing. Apply instant glue at the inserting holes for sealing and dry for 15 min. Then, fill the electrolyte (saturated CuI solution) and insert stoppers to seal the injection holes. Remove any air bubbles present, which can generate a second resonance peak.

Figure 3a shows an exploded view of the sensor. It consists of a 30 mm-long rectangular channel with a 5 mm-wide square cross-section. Its two openings are covered by two Ecoflex membranes, respectively. Figure 3b illustrates the copper and zinc wires inserted into the semi-cylindrical grooves on the bottom wall of the rectangular channel. The glue fills the gap between the electrodes and the wall by capillary forces while the remaining portion is left exposed [23]. It prevents the electrodes from oscillating in the electrolyte during vibration and ensures contact with the electrolyte. Figure 3c illustrates how the glue seals the inserting holes, and Figure 3d shows how the glue underfills the wires with the top portion of the surface left exposed.

### 2.2. Experiment Setup

Referring to Figure 4b, the sensor with the readout circuit and a commercial ADXL 335 reference accelerometer are mounted on a fixture for back-to-back comparison. The fixture is then attached to a vibration exciter, Type 4810, B&K corporation, with their sensing axis (x-axis of ADXL 335) parallel to the movement axis. The scale factor of the ADXL 335 is calibrated by a two-point method using a manual control turntable to obtain the output voltage at ±1 g. The exciter motion instructs by a signal generator and a power amplifier with a voltage gain of 2. Experiments are conducted on a pneumatic isolation table to avoid artifacts.

Figure 5a shows the readout circuit and the sensor’s electrical connection. The zinc and copper wires connect to two RC high-pass filters, each with a cut-off frequency of 1.59 Hz. The filtered outputs then amplify using an instrumentation amplifier AD620 with a differential gain of 413. A 24-bit NI-9202 module records the outputs of the circuit and the reference sensor with a sampling rate of 1 kHz and a 250 Hz low-pass filter. The input into the circuit has a voltage v(t) and a current i(t) component. The galvanic reaction is unidirectional, and i(t) always flows from the copper to the zinc electrode through the two 1 MΩ resistors. A sinusoidal v(t) generates a sinusoidal output without offset (Figure 5b), while i(t) generates a positive sinusoidal output (Figure 5c).

### 2.3. Characterization Procedure

The frequency response, scale factor, long-term response, impedance, and discharge characteristic are characterized. First, the sensors vibrate under a 200 mVpp sine wave between 20–90 Hz to obtain the frequency response. Secondly, they are shaken from ±0.044 g to ±0.580 g under a frequency of 70 Hz for scale factor calibration. Thirdly, they are shaken at ±0.178 g under 70 Hz for 74 min to observe the long-term response. Finally, the electrochemical impedance spectroscopy (EIS) and discharge curves are obtained by Emstat 4s (Palmsens) with a Cu/CuI/Zn cell.

## 3. Results

This section presents the characteristics and analysis of the galvanic vibration sensor. The sensor has a resonance frequency of 68 Hz. Its response is linear from ±0.044 g to ±0.378 g, with a slope of 0.748 V/g and a measurable range of ±0.485 g at 70 Hz. The response magnitude drops to 0.4 times within 74 min due to the spontaneous displacement reaction. The discharge characteristics show that the sensor can operate for a long time with a sufficient electrolyte. 

### 3.1. Frequency Response

The sensor’s frequency response is shown in Figure 6a. The peak sensitivity is 1.152 V/g at the mechanical resonance frequency (68 Hz) and is 178 times more than at 20 Hz. Resonance frequency is a function of electrolyte mass and Ecoflex membrane properties. The displacement of the membrane and the flow rate is maximum at this frequency, thus generating a more significant concentration change and output magnitude.

### 3.2. Scale Factor

Figure 6b plots the magnitude at 70 Hz against acceleration and the regression line with a slope of 0.748 V/g. The response is linear within ±0.044 g and ±0.378 g but deviates after ±0.485 g. The magnitude at ±0.485 g is higher than the regression result and dropped below at ±0.580 g. The sensitivity difference between Figure 6a,b is due to the performance decay, as illustrated in Section 3.4.

### 3.3. Waveform Analysis

The output waveforms in Figure 7 are mostly positive with a slight negative portion. As explained in Figure 5a,b, a voltage input has a bipolar output while a current input has a unipolar output. It indicates that current fluctuation has a significant contribution, and Nernst’s potential variation has a minor contribution. 

Figure 7a shows that the waveform at ±0.485 g under 70 Hz is always sinusoidal. Therefore, it remained inside the measurable range of the sensor. The scale factor above the regression line might be due to the large deformation of the membrane and decreases the effective spring constant.

Figure 7b–d show the waveform at ±0.580 g under 70 Hz at three different time intervals. In Figure 7b, the magnitude at the beginning is similar to that of ±0.485 g. It is the maximum amplitude of the sensor in the current electrochemical and mechanical conditions. In Figure 7c, the magnitude decreases after some time and indicates a shortage of ions near the electrode. The reason could be that the strong convection prevents ions diffusing towards the electrodes. Finally, in Figure 7d, the waveform is distorted and no longer sinusoidal after 8 s.

### 3.4. Long Term Response

Referring to Figure 8a, the DC component at various accelerations within ±0.485 g is similar. It indicates that the lifetime of the sensor has no significant correlation with the vibration strength within the measurable range. The input offset of AD620 is 125 μV max and the output offset is 1.5 mV max, which are both smaller than the above data. Thus, the data can genuinely represent the offset due to the sensor.

Figure 8b shows the output magnitude calculated from the Fourier transform with a series of consecutive 30 s windows under an acceleration of ±0.179 g at 70 Hz for 74 min. It shows that the response magnitude drops to 0.4 times of the initial within 74 min. The data fit with an equation V=8.632/(t+51.4) and shows the performance decay with 1/t. The initial three data points are taken under vibration and stopped. The trend is consistent with the data points later on under continuous vibration. It indicates that longevity does not correlate with vibration and sensor output, supporting the data in Figure 8a. The reason could be that the average reaction rate is dependent on the electrochemical system only.

The spontaneous displacement reaction occurs irrelevant to vibration. Vibration allows the direct current to modulate into the sinusoidal signal as an offset and read out by the circuit. Therefore, the high-pass filter only blocks the direct current in the absence of vibration, but filtering does not affect the lifetime significantly.

### 3.5. Electrochemical Impedance Spectroscopy (EIS) and Discharge Characteristics

Figure 9 shows the result of EIS with an amplitude of 10 mV, centered at the open circuit potential from 0.05 Hz to 100 kHz. The Nyquist plot (Figure 9a) does not offer a typical hemisphere and cannot well fit the Randles equivalent circuit. The magnitude (Figure 9b) at high frequency (>100 Hz) for all conditions and at low frequency (<100 Hz) before stirring are similar. Before stirring, the phase plot (Figure 9c) has a similar shape but is significantly different after. 

Figure 10 shows the discharge characteristics at various rates (a) and discharge under 0.9 μA for one hour (b). The cell can discharge stably with sufficient electrolyte volume for a long time, unlike the decay in sensor performance in Figure 8b. Therefore, the consumption of electrode material is small due to the low solubility of CuI in water. The sensor performance can restore after electrolyte refilling.

## 4. Conclusions

In summary, this paper reports a novel galvanic vibration sensor that integrates an energy storage function and a sensing function. The sensor consists of a simple galvanic cell with zinc, copper, and copper iodide electrolytes in a rectangular channel. The channel is sealed with elastic membranes to form a mass–spring system. The frequency response, scale factor, waveform, long-term performance, impedance, and discharge characteristics are presented. The sensor has a resonance frequency of 68 Hz, a scale factor of 0.748 V/g at 70 Hz, and a maximum measurable range of ±0.485 g. Due to the spontaneous displacement reaction, the sensor sensitivity performance drops to 0.4 times within 74 min but can restore after refilling the electrolyte. 

The lifetime of the sensor can be improved by: 1. placing a porous separator between the copper and zinc electrode to reduce bulk mixing while enabling fluid flow; 2. injecting two different electrolytes that contain a high concentration of zinc and copper ions into the two compartments containing the zinc and copper electrode respectively; 3. Increasing electrolyte volume. Future work can replace the Zn-Cu cell with a rechargeable one such as a lead–acid cell and include a sensor-powered circuit. Self-powered vibration sensors are essential in large-scale vibration monitoring networks such as structural health monitoring, machine health monitoring, and aircraft health monitoring.

## Figures and Tables

**Figure 1 micromachines-13-00530-f001:**
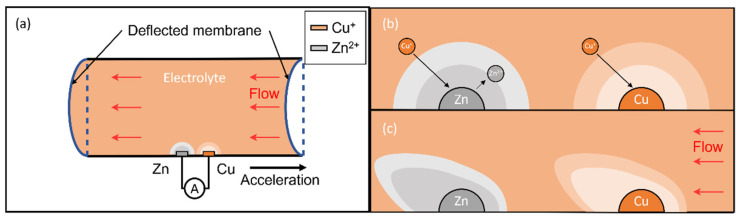
Illustration of the galvanic vibration sensor mechanism. (**a**) Schematic illustration of the sensor under acceleration. Bulk electrolyte moves opposite to acceleration and stops by the elastic membranes. The current from the Zn and copper electrodes passing through an external varies with the flow generated. (**b**) Drawing of the diffusion layer formed due to spontaneous displacement reaction and the ion diffusion without acceleration. (**c**) Drawing of the diffusion layer with acceleration.

**Figure 2 micromachines-13-00530-f002:**
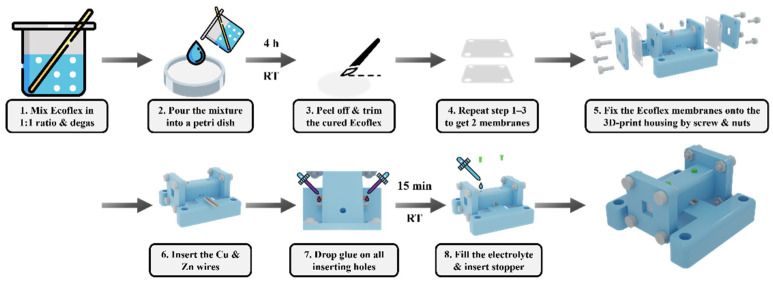
The fabrication procedure of the Ecoflex membranes (step 1–3) and the galvanic vibration sensor (4–8).

**Figure 3 micromachines-13-00530-f003:**
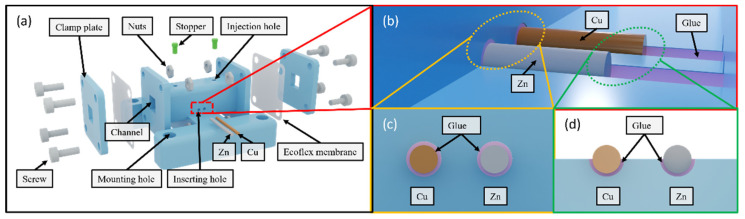
(**a**) The exploded view of the sensor; (**b**) the illustration of the copper and zinc wires in the semi-cylindrical grooves inside the rectangular channel; (**c**) the illustration of glue surrounding the metal wires at the inserting holes; (**d**) the illustration of glue underfilling the metal wires inside the rectangular channel.

**Figure 4 micromachines-13-00530-f004:**
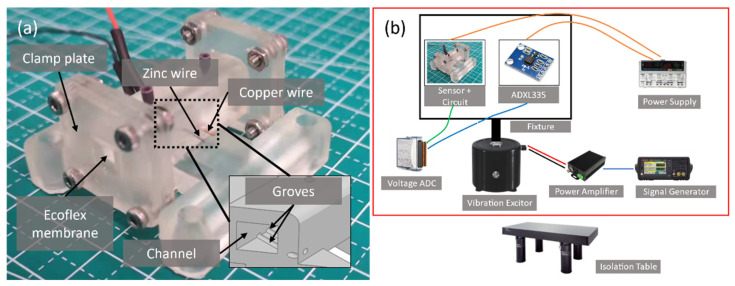
(**a**) A photograph of the fabricated sensor. (**b**) Showing the experimental setup of the vibration sensor characterization. A vibration exciter moves the sensor and the ADXL 335 reference accelerometer sinusoidally according to the signal generator output. The output from the sensors is recorded using a voltage acquisition card.

**Figure 5 micromachines-13-00530-f005:**
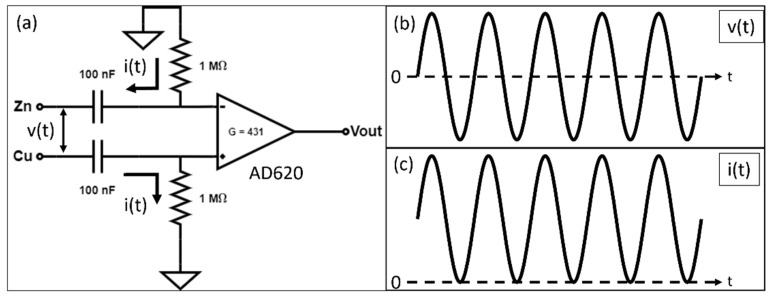
(**a**) The readout circuit of the galvanic vibration sensor. The zinc and copper wires connect to two RC high-pass filters with a cut-off frequency of 1.59 Hz. The filtered outputs then amplify using an instrumentation amplifier AD620 with a differential gain of 413. The voltage input v(t) and current input i(t) from the vibration sensor have different output behaviors. (**b**) The output behavior of v(t) with no offset voltage in the output; (**c**) the i(t) output behavior with an offset voltage, and always positive.

**Figure 6 micromachines-13-00530-f006:**
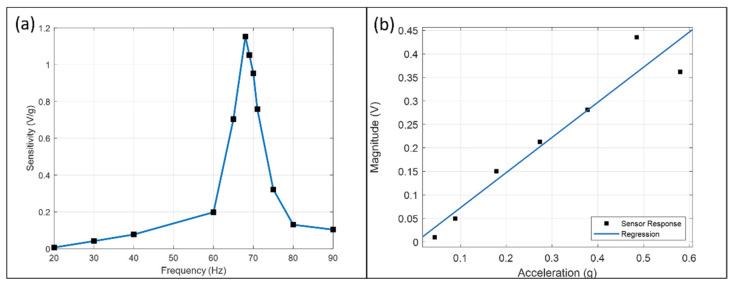
(**a**) The frequency response of the galvanic vibration sensor, plotting the scale factor as a function of frequency in linear scale from 20 Hz to 90 Hz. It peaks at the mechanical resonance frequency of the sensor (68 Hz). (**b**) The signal magnitude of the vibration sensor against acceleration from ±0.044 g to ±0.580 g and the regression line. The regression agrees from ±0.044 g to ±0.378 g with a slope of 0.748 V/g.

**Figure 7 micromachines-13-00530-f007:**
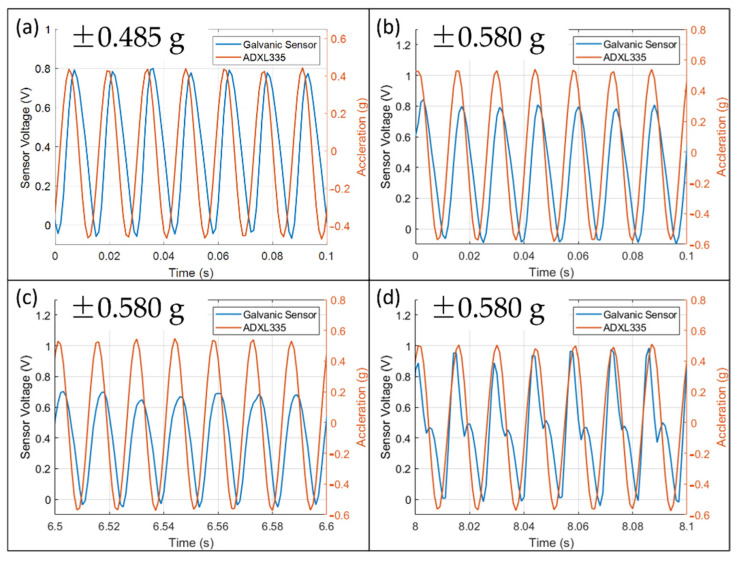
The output waveform of the galvanic vibration sensor. (**a**) The output waveform of the galvanic vibration sensor is at ±0.485 g, is sinusoidal and stable. It indicates it is within the measurement range of the sensor. (**b**–**d**) Output waveform under ±0.580 g at 70 Hz, (**b**) t = 0 s to 0.1 s; (**c**) t = 6.5 s to 6.6 s; (**d**) t = 8 s to 8.1 s. The waveform is unstable when the acceleration is outside the sensor’s measurement range.

**Figure 8 micromachines-13-00530-f008:**
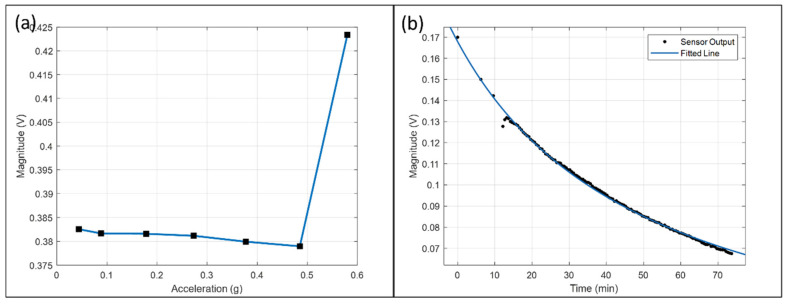
(**a**) The DC component of the waveform against acceleration; (**b**) The output magnitude over time under ±0.179 g at 70 Hz for 74 min.

**Figure 9 micromachines-13-00530-f009:**
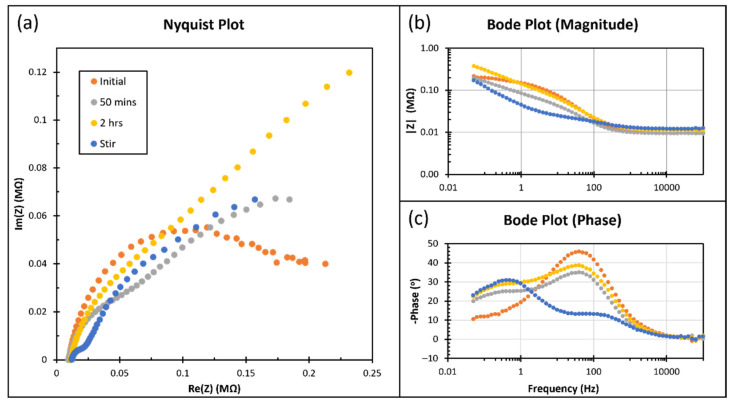
Electrochemical impedance spectroscopy of a Cu/CuI/Zn cell with 10 mV AC voltage and open circuit potential as DC voltage from 0.05 Hz to 100k Hz. The data obtained when: 1. Electrodes just immersed in the electrolyte; 2. After 50 min; 3. After two hours; 4. After being immersed for two hours and finished stirring. (**a**) Nyquist plot. (**b**) Magnitude part of Bode plot; (**c**) Phase part of Bode plot.

**Figure 10 micromachines-13-00530-f010:**
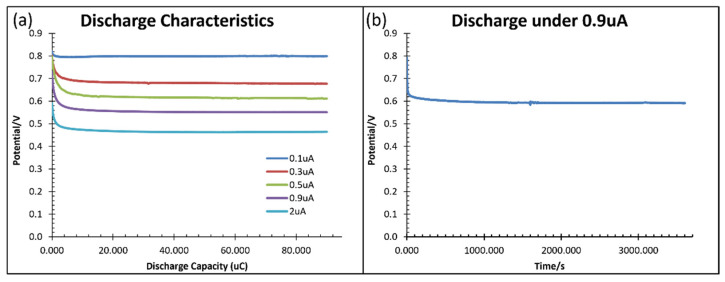
Discharge characteristics of the Cu/CuI/Zn cell with a large amount of saturated CuI solution. (**a**) At a discharge rate from 0.1–2μA; (**b**) Discharge under 0.9 μA for one hour without significant degradation.

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
