# Peer review of "Self-Powered Galvanic Vibration Sensor"

_micromachines, 2022, doi:10.3390/mi13040530_

Round 1

Reviewer 1 Report

The manuscript titled “self-powered galvanic vibration sensor” by Cheung et al focus on the energy harvester and energy storage device integration. The performance of the sensor is more than halved within 74 minutes. The longevity of the sensor is short due to the spontaneous reaction and can be improved by altering the composition of the galvanic cell. After careful evaluation, a major revision needs to be done before a final decision. My comments are as follows:

  1. In the introduction add some of the latest self-charging papers and discuss the importance of different self charging process: (a) Sustainable Materials and Technologies, 32, e00396,2022 and (b) Nano Energy 89, 106316, 2021

  1. In section 2.1, it is better to add the illustration regarding the fabrication of the sensor for a better understanding of the readers.

  1. Why did the authors choose the MEMS accelerometer ADXL 335 as a reference even though a lot of commercial sensors are available in the market? Is there really any fascinating about the ADXL 335? Please explain in detail.

  1. The authors did not show any type of impedance study for their galvanic cell

  1. The long-term stability of the device like the charging-discharging curve should be investigated?

  1. In conclusion, the authors should add some important areas of application for the fabricated galvanic cell.

  1. In this paper, the author needs to give a great deal of explanation about the general mechanism of their device?

  1. English needs to be polished and grammatical errors need to be removed carefully.

Reviewer 2 Report

In general, this work is interesting and it is worth publishing after a major revision process, four questions are listed below and I hope the author can further explain to us.

Questions:

  1. I think the authors need to provide more information about the structure of the galvanic sensor, especially more detailed photos because I found the vibration sensor’s structure in figure 1(b) is a little bit confusing. Too small to figure out which parts are corresponded to figure 1(a). And in the fabrication of the sensor, I think the authors need to use figures to illustrate the process.
  2. The output waveform could only keep a sinusoidal shape for about 8 seconds, is it possible that this phenomenon can be contributed to the displacement reaction? Will it be helpful if the authors change the type of electrolyte?
  3. The performance of the sensor was not stable in less than 10 seconds, for sensors, this performance is far from satisfactory, the author should try to use the methods they mentioned to improve the stability.
  4. Again, I am very interested in the detailed structure of the sensor, is this sensor disposable? Can the author replace the electrolyte and the two electrodes when the sensor’s performance is not as good as it used to be?

Round 2

Reviewer 2 Report

Thanks for the authors's reply, I think all of my questions have been answered. However, it will be much more intuitive if the authors are willing to provide a video which the sensor is working and signals are being readout.